# Optimization of Coherent Dynamics of Localized Surface Plasmons in Gold and Silver Nanospheres; Large Size Effects

**DOI:** 10.3390/ma16051801

**Published:** 2023-02-22

**Authors:** Krystyna Kolwas

**Affiliations:** Institute of Physics, Polish Academy of Sciences, Al. Lotników 32/46, 02-668 Warsaw, Poland; krystyna.kolwas@ifpan.edu.pl

**Keywords:** Localized Surface Plasmons (LSP), plasmon damping, coherence dephasing, dispersion relation, open quantum system, quasi-particle, Au nanoparticles, Ag nanoparticles, size effects, quality factor

## Abstract

Noble metal nanoparticles have attracted attention in recent years due to a number of their exciting applications in plasmonic applications, e.g., in sensing, high-gain antennas, structural colour printing, solar energy management, nanoscale lasing, and biomedicines. The report embraces the electromagnetic description of inherent properties of spherical nanoparticles, which enable resonant excitation of Localized Surface Plasmons (defined as collective excitations of free electrons), and the complementary model in which plasmonic nanoparticles are treated as quantum quasi-particles with discrete electronic energy levels. A quantum picture including plasmon damping processes due to the irreversible coupling to the environment enables us to distinguish between the dephasing of coherent electron motion and the decay of populations of electronic states. Using the link between classical EM and the quantum picture, the explicit dependence of the population and coherence damping rates as a function of NP size is given. Contrary to the usual expectations, such dependence for Au and Ag NPs is not a monotonically growing function, which provides a new perspective for tailoring plasmonic properties in larger-sized nanoparticles, which are still hardly available experimentally. The practical tools for comparing the plasmonic performance of gold and silver nanoparticles of the same radii in an extensive range of sizes are also given.

## 1. Introduction

The plasmonic properties of nanoscale structures lay the groundwork for many future technologies, applications, and materials (e.g., [1,2,3,4,5,6,7,8]). Plasmonics exploit the unique properties of metal/dielectric interfaces [9,10,11,12,13,14,15,16,17,18,19] and references therein), which enable the concentration manipulation of electromagnetic (EM) fields on the subwavelength scale. Plasmonic nanostructures provide ways to generate, confine, guide, modulate and detect light.

Metal-dielectric interfaces support surface plasmons—collective surface charge density oscillations of free electrons [15,20,21,22]. In the case of finite-size nanostructures, such oscillations form the standing waves of Localised Surface Plasmons (LSP) [15,23,24], which are damped as a result of basically different physical mechanisms.

In spherical metal nanoparticles (MNPs), the basic plasmonic properties can be manipulated by the radius. The contribution of LSPs, which is resonant in character, is present in the scattering, absorption, or extinction spectra and, if dominant, manifests in the form of maxima with a size-dependent spectral position, spectral width, and amplitude. So the maxima in the far-field spectra reflect the resonant character of the EM excitations, which arise at the metal-dielectric interface and are direct consequences of the size-dependent intrinsic properties of plasmonic nanoparticles (NPs) [15,25] itself. In optical research, the scattering and absorption spectra can be predicted using the Lorentz–Mie scattering theory applied to MNPs of a chosen radius. However, Mie theory does not give direct information either about the size-dependent pick positions in the spectra, the LSP resonance condition or the basis of plasmon damping processes reflected in the spectral width of the maxima and their amplitudes.

So, in many optical issues related to LSPs, it is not the behaviour of the electrons that is of primary interest, but the surface-localized EM fields coupled to charge oscillations and confined to the metal–dielectric interface. In resonance, the incoming EM waves are effectively captured at the spherical metal/dielectric interface forming 3D standing waves coupled to oscillations of the nanoparticle’s conduction band electrons.

LSP resonance frequencies and damping rates in the function of particle size are the core parameters in applications as size-dependent plasmon dynamics is a fundamental tool in applications. In particular, understanding the dephasing of coherent electron oscillation and its consequences on associated EM fields is crucial. The sketchily called $T_{1}$ and $T_{2}$ times introduced in analogy to the widely used phenomenological formula from NMR spectroscopy used e.g., in [26,27,28,29,30,31] are in general not satisfactory for the case of plasmons. Therefore, the more general description of the multipolar plasmon times (rates) seems to be interesting.

The first part of the present report contains a recapitulation of the EM description, which includes the LSP dynamics resulting from considering the dispersion relation (DR) for Surface Localized EM (SLEM) fields in gold and silver nanoparticles. The noble metal nanoparticles have been used in different applications such as biosensing, catalysis, pollutant degradation, solar cells, and hydrogen production (e.g., [7,12,32,33,34,35,36]). Experimental studies [25,37,38,39] and applications of the noble MNPs are limited by the sizes of NPs due to production technology problems. Large sodium droplets [40] induced from the vapour phase make an exception, which is, however, not interesting for solid-state plasmonics. In many basic studies, the commercially available, highly monodisperse particles of limited size are used. Theoretical studies need not be constrained by similar problems, and that allows us to promote the interest in plasmonic properties of relatively large MNPs, still hardly available experimentally.

After a brief summary of the classical EM cavity surface mode description based on DR for SLEM modes [15,41,42], we recall results obtained from the model, in which the plasmonic system is treated in terms of atomic quasi-particles (QP) of energy levels corresponding to the energies of plasmonic mode oscillations. In particular, using the QP picture and Lindblad (e.g., [43,44]) equations for open systems, we distinguish between decoherence damping, which affects the relative phases, and population relaxation, which affects the number of electrons involved in the coherent motion in the processes of the radiative and nonradiative damping experienced by LSPs.

Linking classical and quantum pictures, we provide the size dependence of the decoherence and the depopulation damping rates of higher than a dipole, multipolar plasmons, in the NPs size range not limited by approximations. It allows us to discuss the quality factors of plasmonic cavities and to demonstrate the high efficiency of plasmonic performance of nanoparticles in size ranges, which are still not available experimentally.

## 2. Classical Picture Based on Maxwell’s Equations

### 2.1. Mie Scattering Theory versus Dispersion Relation (DR) for Surface Localized EM Fields

Mie scattering theory answers the question of how the EM field of the incident plane wave is modified by the presence of a homogeneous sphere of a chosen radius [45] in a non-absorbing dielectric medium. After applying the appropriate continuity relation at the sphere’s boundary, the distribution of the EM fields inside and outside the sphere can be found [46]. Then, e.g., the total cross-sections for absorption, scattering, and extinction (and the corresponding efficiencies (e.g., [15])) or the far-field spectra (which display maxima in size-dependent spectral ranges) can be obtained (see Figure 1a). Such maxima are manifestations of plasmon resonances.

However, in Mie scattering formalism, the resonance condition and its size dependence are not specified. Additionally, there is no direct information about the decay processes nor the size dependence of the rates of such processes. Answers to these questions are hidden in the intrinsic characteristics of a plasmonic particle alone (Figure 1b) and manifest in the measured quantities, when the particle is illuminated.

### 2.2. Dispersion Relation and the Resulting Oscillation Energies and Damping Rates of Plasmonic Cavity Modes Versus Metal NP’s Radius

Dispersion relation (DR) connects spatial and temporal characteristics of the waves, which can propagate in a medium. In the elementary case of harmonic waves in the bulk dielectric medium, the DR connects the oscillation frequencies to the wave numbers of waves, which can propagate in the medium.

When looking for the DR for plasmonic SLEM fields at a spherical metal-dielectric interface, one considers a formal scheme similar to that of the Mie scattering theory (Figure 1a), but in the absence of the field from outer sources (Figure 1b) [12,15,41,47,48,49]. Such a scheme, in the case of the TM (Transverse Magnetic) component of EM field (see Figure 2) is analogous to that for a flat metal–dielectric interface, which describes the propagation of Surface Plasmon Polariton (SPP) (e.g., [15]). In both cases, the component of the electric field normal to the interface is present and plays a crucial role. However, in the case of a spherical interface, the dispersion relation is formally more complicated:(1)$$\sqrt{\varepsilon_{in}}\xi_{l}^{\prime }\left(k_{out}R\right)\psi_{l}\left(k_{in}R\right)-\sqrt{\varepsilon_{out}}\xi_{l}\left(k_{out}R\right)\psi_{l}^{\prime }\left(k_{in}R\right)=0$$
as it contains the special functions of complex arguments and the derivatives of these functions with respect to these arguments. For the transverse electric (TE) component of the field, the corresponding dispersion relations have no solutions for Reεin(ω,R)<0. The complex $\psi_{l}\left(z\right),$
$\xi_{l}\left(z\right)$ are Riccati–Bessel spherical functions (of complex arguments), which can be expressed by the Bessel $J_{l+1/2}\left(z\right)$, Hankel $H_{l+1/2}^{\left(1\right)}\left(z\right)$, and Neuman $N_{l+1/2}\left(z\right)$ cylindrical functions of the half-order (Let us note, that in some ready-to-use numerical procedures dealing with special functions it is assumed, that the arguments are real. However, in the numerical search of conditions for the existence of solutions of the DR (Equation (Equation 1)) the complexity of the arguments must be taken into account). $k_{in}$ and $k_{out}$ are the wave vectors inside the sphere, and in the sphere surroundings, respectively; $k_{in}\left(\omega ,R\right)=\sqrt{\varepsilon_{in}\left(\omega ,R\right)}\cdot \omega /c$, and $k_{out}\left(\omega \right)=\sqrt{\varepsilon_{out}}\cdot \omega /c$ are the dispersion relations in the extended media inside and outside the sphere; $\varepsilon_{in}\left(\omega ,R\right)$ and $\varepsilon_{out}$ are the dielectric functions of the metal sphere and of the dielectric non-absorbing environment, respectively, and *c* is the speed of light. The dielectric function of the metal NP $\varepsilon_{in}\left(\omega ,R\right)$ includes the size-dependent effect of an additional electron relaxation due to collisions with the sphere boundary (taken into account phenomenologically) and the impact of the interband electronic transitions (see [15]). All the numerical results below are obtained for gold and silver NPs in immersion ($n_{out}=\sqrt{\varepsilon_{out}}=$1.5).

The dispersion relation (Equation (Equation 1)) results from the continuity relations at the sphere boundary imposed on the solutions of the self-consistent, divergent-free Maxwell equations in the absence of the incoming light field [47] (recollected in [48]), according to the scheme presented in Figure 2. The solutions exist only for the complex, discrete eigenvalues $\omega_{l}\left(R\right)-i2\Gamma_{l}\left(R\right)$ (In our previous papers (see e.g., [15] and references therein) we have used $2\Gamma_{l}\left(R\right)$ for denoting the imaginary part of the complex eigenvalues. Such a choice has been motivated by connecting $\Gamma_{l}$ to the Lorentzian profile of LSP resonances manifesting in the spectra in the case of small damping ($\Gamma_{l}\ll \omega_{l}$) for the normalized profile such that the central intensity of the mode *l* $I\left(\omega_{l}\right)=I_{0}$. In such a case, the Lorentzian profile $I\left(\omega \right)=I_{0}\frac{\Gamma_{l}^{2}}{{\left(\omega -\omega_{l}\right)}^{2}+\Gamma_{l}^{2}}$ with $2\Gamma_{l}$ denoting the full width at half maximum (FWHM) of the profile [50]). Calculated for successive radii *R*, it allows us to find the radius-dependent oscillation dynamics of the mode *l* of the surface localized harmonic EM field $E_{l}^{SLEM}\left(r=R,\theta ,\phi ,t\right)$ allowed at the interface:(2)$$E_{l}^{SLEM}\left(r=R,\theta ,\phi ,t\right)\propto \exp\left(\left(i\omega_{l}\left(R\right)-2\Gamma_{l}\left(R\right)\right)t\right)$$

Such oscillation dynamics are unambiguously determined by the intrinsic properties of the NP of a given size and its dielectric environment. The parameters $\omega_{l}$ and $\Gamma_{l}$ characterise the plasmonic NP itself, disregarding whether the particle is illuminated or not. The excitation of LSP is a resonance process, which takes place when the frequency of the incoming light ω fits the eigenfrequency (-ies) of a plasmonic resonator $\omega_{l}\left(R\right)$ (of the cavity mode(s) *l*), with *l* = 1,2,3, ….

However, as far as the role of $\omega_{l}$ is rather clear, the role of the damping rates $\Gamma_{l}$ needs further clarification. For this purpose, one can use an image, in which the plasmonic particle is described as a quasi-particle [42].

## 3. The Plasmonic Quasi-Particle Decay Dynamics

The pairs of $\omega_{l}\left(R\right)$ and $\Gamma_{l}\left(R\right)$, found in absence of the illuminating radiation, characterize an NP of the radius *R* in the same way as the energy levels and the inverse of lifetimes characterize an atom or a molecule. In both cases, these quantities manifest in the spectra, when the systems are illuminated.

Let us ascribe [42] the oscillation energies $\omega_{l}$ of the classical EM modes of the plasmonic cavity of radius *R* to the discrete energy levels of electronic states, which are higher than the zero-energy non-oscillatory level by the energies $\hbar \omega_{l}\left(R\right)$ (see Figure 3). The corresponding states of the plasmonic system *S* in the Hilbert space are |l〉 with *l* = 1, 2, 3…. The only possible transitions are those with the absorption or emission of a photon with the energy $\hbar \omega_{l}$. Such transitions occur between the states |l〉 and the non-oscillatory state |0〉. *N* electrons of the plasmonic system *S* can be, in general, distributed over the states |n〉, n=0,l =1, 2… if the system was initially excited accordingly. If the system is not excited, all electrons remain in the ground state |0〉, which is the lower energy, stationary state of the system *S*.

### 3.1. The Density Matrix Operator and Quantum Master Equation

To describe the state of the plasmonic system S and its evolution, we use the density matrix formalism, which is convenient for describing the quantum systems in mixed states and for time-dependent problems. The diagonal elements of the density matrix correspond to the probabilities $p_{n}$ of occupying quantum states |n〉, which are proportional to the relative electron populations $N_{n}/N$ of these states. The complex off-diagonal elements describe quantum coherences, which contain time-dependent phase factors. In time-dependent problems, the off-diagonal elements describe the evolution of the coherent superposition of these states.

As no physical system is absolutely isolated from its surroundings, the plasmonic system *S* has to be considered an open quantum system, which is a subsystem of a larger combined quantum system S+E, where *E* represents the environment to which the open system *S* is coupled. Following the main assumption of the basic theory of open quantum systems (e.g., [51,52]), the environment is assumed to be a large system with an infinite number of degrees of freedom. The interaction of the open system *S* with the environment is assumed to cause an irreversible behaviour for the open system *S* and leads to decoherence (randomization of phases) and dissipation of energy into the surroundings. The evolution of *S* can be described by a quantum Markovian master equation.

The system *S* can be described as a sum of independent, open, two-level subsystems $S_{l}$ [42]: $S=\sum_{l=1}S_{l}$ under the assumption that the states |l〉 (of energy corresponding to the self-frequency of the cavity modes) are not coupled. Each two-level sub-system (Figure 4a) consists of: the excited |l〉 state and the ground, non-oscillatory state |0〉, which is the lowest energy, stationary state. The dynamics of the system *S* thus result from the independent dynamics of the systems $S_{l}$, which can be described following the scheme of a standard textbook 2-level systems applied to many model physical systems. In the basis of states:(3)$$\;|l\rangle =\left(\begin{array}{c} 1\\ 0 \end{array}\right),\;|0\rangle =\left(\begin{array}{c} 0\\ 1 \end{array}\right),$$
the density-matrix operator $\rho^{S_{l}}\left(t\right)$ is represented by a 2 × 2 matrix:(4)$$\rho^{S_{l}}\left(t\right)=\left(\begin{array}{cc} \rho_{ll}\left(t\right) & \rho_{l0}\left(t\right)\\ \rho_{0l}\left(t\right) & \rho_{00}\left(t\right) \end{array}\right).$$

### 3.2. The Hamiltonian of the Uncoupled System

Quantum properties of the uncoupled EM cavity modes *l* are carried by the mode “amplitudes”: annihilation $a_{l}\;$ and creation $a_{l}^{+}$ operators of photons, which satisfy: $\left[a_{l},a_{l}^{+}\right]=a_{l}a_{l}^{+}-a_{l}a_{l}^{+}=1$.

A quantisation of the classical field Hamiltonian (e.g., [53]) gives the Hamilton operator of the EM field: $H_{l}^{F}=\left(\hbar \omega_{l}/2\right)\left(a_{l}a_{l}^{+}+a_{l}^{+}a_{l}\right)=\hbar \omega_{l}\left(a_{l}a_{l}^{+}-1/2\right).$ By redefining the zero energy level, one can drop 1/2 in the *l*-th field Hamiltonian: $H_{l}^{F}=\hbar \omega_{l}a_{l}a_{l}^{+}$. The additional argument for dropping 1/2 is that the constant energy in the Hamiltonian commutes with *a* and $a^{+}$ so it cannot affect the quantum dynamics described by the Heisenberg equations of motion.

In the picture of the plasmonic QP, we introduce the Hamiltonian $H_{l}$, phenomenologically equivalent to $H_{l}^{F}$, in the form [42]: $H_{l}=\hbar \omega_{l}\sigma_{+}\sigma_{-}$, where $\sigma_{-}$ and $\sigma_{+}$ are the energy-lowering (a photon is created) and energy-rising operators (a photon is annihilated) (see Figure 4a). In the isolated (closed) system $S_{l}$ (no damping processes) the system $S_{l}$ is in the state of the initial coherent superposition of states caused by absorption and annihilation of photons from and into the mode *l*, which in the model of plasmonic QP in a self-cavity is equivalent to energy loss and gain at an amount $\hbar \omega_{l}$, as sketched in Figure 4a. Phenomenological association of the Hamiltonians $H_{l}^{F}$ and $H_{l}$ enables a standard description of the losses suffered by the quantum system $S_{l}$ (see next Subsection). Let us note for clarity, that the system described here is much simpler than that in the case of the Jaynes–Cummings (e.g., [54]) model and its extensions.

### 3.3. Relaxation Processes

An excited plasmon, like an excited atom, decays to a state of lower energy spontaneously emitting a photon. In the theory of open quantum systems, such decay is assumed to be due to the coupling of the system to the environment (see Figure 4b). Such coupling introduces the radiative losses via spontaneous decay (coupling to the EM vacuum fluctuating fields) and heat losses due to the inevitable collisions of electrons in metal (coupling to a heat bath in thermal equilibrium). The dynamics of both coupling processes are assumed to be much faster than those of the open system $S_{l}$, so the dynamics of $S_{l}$ (and that of *S*) are Markovian.

Each system $S_{l}$ is assumed to be coupled to the environment independently. In such a case, the general form of the Lindblad equation (e.g., [43,44]) also guarantees that the dynamics of each matrix operator $\rho^{S_{l}}$ describing $S_{l}$ are governed by the Lindblad equation in the form:(5)$$\frac{\partial \rho^{S_{l}}\left(t\right)}{\partial t}=-\frac{i}{\hbar }\left[H_{l},\rho^{S_{l}}\left(t\right)\right]-\frac{1}{2}\sum_{\alpha =r,nr}\left(L_{\alpha ,l}^{†}L_{\alpha ,l}\rho^{S_{l}}+\rho^{S_{l}}L_{\alpha ,l}^{†}L_{\alpha ,l}-2L_{\alpha ,l}\rho^{S_{l}}L_{\alpha ,l}^{†}\right)$$

The first term on the right-hand side of Equation (Equation 5) describes the unitary evolution of the two-level subsystem $S_{l}$ under the Hamiltonian $H_{l}=\hbar \omega_{l}\sigma_{+}\sigma_{-}$. In the second, dissipative term (the so-called Lindblad dissipator), the summation over α extends over all processes (radiative and nonradiative) of coupling the system $S_{l}$ to the environment. The Lindblad dissipator contains the “jump” operators $L_{\alpha ,l}$ expressed by the energy-lowering operators:(6)$$L_{s,l}=\sqrt{\Gamma_{l}^{s}}\sigma_{-},\;L_{col,l}=\sqrt{\Gamma_{l}^{col}}\sigma_{-}$$
which describes the random, sudden emission of a photon from the state |l〉 to the state |0〉. The radiative and nonradiative rates $\Gamma_{l}^{r}=\Gamma_{l}^{s}$ and $\Gamma_{l}^{nr}=\Gamma_{l}^{col}$ are the rates of damping processes resulting from irreversible coupling to the outer radiation and heat reservoirs (see Figure 4b). $\sigma_{-}=|0\rangle \langle l|$ is the energy lowering operator, $\sigma_{+}=\sigma_{-}^{†}=|1\rangle \langle 0|$, is the energy rising operator. $\sigma_{-}$ and $\sigma_{+}$ operators can be expressed by means of 2 × 2 Pauli matrices $\sigma_{1}$ and $\sigma_{2}$: $\sigma_{-}=\left(\sigma_{1}-i\sigma_{2}\right)/2,\sigma_{+}=\left(\sigma_{1}+i\sigma_{2}\right)/2$.

After algebra involving 2 × 2 matrices, the Lindblad master Equation (Equation 5) including radiative and nonradiative dissipation processes (e.g., [25]) allows finding the evolution of populations of the excited and ground states:(7)$$\begin{array}{ccc} \rho_{ll}\left(t\right) & = & \rho_{ll}\left(t_{0}\right)\exp\left(-\Gamma_{l}^{tot}\left(t-t_{0}\right)\right), \end{array}$$(8)$$\begin{array}{ccc} \rho_{00}\left(t\right) & = & \rho_{ll}\left(t_{0}\right)\left(1-\exp\left(-\Gamma_{l}^{tot}\left(t-t_{0}\right)\right)\right)+\rho_{00}\left(t_{0}\right) \end{array}$$
and of coherences:(9)$$\begin{array}{ccc} \rho_{l0}\left(t\right) & = & \rho_{0l}\left(t_{0}\right)\exp\left(i\omega_{l}-\Gamma_{l}^{tot}/2)\left(t-t_{0}\right)\right)=\rho_{0l}^{*}\left(t\right), \end{array}$$
where $\Gamma_{l}^{tot}=\Gamma_{l}^{s}+\Gamma_{l}^{col}$. So the dephasing of coherences $\Gamma_{l}^{coh}=\Gamma_{l}^{tot}/2$ is twice as slow as the damping of populations: $\Gamma_{l}^{pop}=\Gamma_{l}^{tot}$.

## 4. Linking the Results of the Classical EM and Quantum Modelling

### 4.1. Size Dependence of the Damping Rates of Coherences and Population

Classical modelling, based on the DR, allows finding the intrinsic size-dependent oscillation frequencies $\omega_{l}\left(R\right)$ of harmonic waves at the interface and damping rates $2\Gamma_{l}\left(R\right)$ of these oscillations as the function of the NP’s radius *R*. The resulting damping of oscillation of the SLEM fields $E_{l}^{SLEM}\left(r=R,\theta ,\phi ,t\right)$: $E_{l}^{SLEM}\left(t\right)/E_{0,l}\left(r=R,\theta ,\phi ,t\right)=\exp\left(i\omega_{l}-2\Gamma_{l}\right)\left(t\right)$ corresponds to the dynamics of coherences $\exp\left(i\omega_{l}-\Gamma_{l}^{coh})\right)\left(t-t_{0}\right)$ (Equation (Equation 9)) in the quantum QP modelling, which describes the decoherence (dephasing of electron oscillations) process at the rate of $\Gamma_{l}^{coh}$ (Equation (Equation 9)). Linking the quantum and classical picture, one can get the size dependence of the rates of decoherence, as we know the dependence on the size of the classical quantity $\Gamma_{l}\left(R\right)$. Thus, $\Gamma_{l}^{coh}\left(R\right)=2\Gamma_{l}\left(R\right)$.

The diagonal elements of the density matrix (Equation (8)), which are proportional to relative populations of the excited state $\rho_{ll}^{S_{l}}\left(t\right)=N_{l}\left(t\right)/N$, correspond to amplitudes $E_{0,l}\left(t\right)$ of the SLEM field $E_{l}^{SLEM}$ (Equation (Equation 2)) in the classical modelling. Therefore, in a free-evolving plasmonic system, we can ascribe size dependence to $\Gamma_{l}^{pop}$: $\Gamma_{l}^{pop}\left(R\right)=\Gamma_{l}\left(R\right)$.

The resulting size dependence of the coherence and population damping rates $\Gamma_{l}^{coh}\left(R\right)$ and $\Gamma_{l}^{pop}\left(R\right)$ for gold and silver NPs are presented in Figure 5.

### 4.2. Transient Decay Dynamics of a Plasmonic System

The example of the SLEM field dynamics resulting from the dephasing of electron motion with the rate of $\Gamma_{l=1}^{coh}\left(R\right)$ is presented in Figure 6 for the case when only the dipole plasmon mode in an NP of the radius R=10 nm was initially excited (or equivalently, only the state |l=1〉 of a QP was initially populated).

The transient decay dynamics of the total plasmonic SLEM field:(10)$$E_{l}^{SLEM}\left(t\right)=E_{0,l}\left(t=t_{0}\right)\exp\left(-\Gamma_{l}^{pop}\right)\exp\left(i\omega_{l}-\Gamma_{l}^{coh}\right)\left(t\right)$$
in the dipole mode is presented in Figure 6b.

In general, the damping of the total SLEM field in the successive *l* modes takes place with the rates: $\Gamma_{l}^{pop}+\Gamma_{l}^{coh}=\frac{3}{2}\Gamma_{l}^{pop}=3\Gamma_{l}^{coh}=\Gamma_{l}$.

## 5. Quality Factors of LSP Cavity

The quality factors of the plasmonic nano-cavity, which we discuss, are based on the definition of a quality factor of a resonator defined as the ratio of the initial energy stored to the energy lost in one radian of the cycle of oscillation. For EM fields with time dependence: $E\propto \exp\left(-\alpha t\right)\cos\left(\omega t+\phi_{1}\right)$ and $H\propto \exp\left(-\alpha t\right)\cos\left(\omega t+\phi_{2}\right)$ the EM energy is proportional to $\frac{1}{4}{\epsilon |E|}^{2}+\frac{1}{4}\mu {\left|H\right|}^{2}$, so the time-average of EM energy is: $\langle W\rangle =\langle W_{E}\rangle +\langle W_{h}\rangle =W_{0}e^{\left(-2\alpha t\right)}$. The average stored energy decreases to 1/e value of its initial value at t=τ=1/2α. Therefore, the quality factor is Q=ω/2α.

The quality factor $Q_{l}^{coh}\left(R\right)$ of plasmonic nano-cavity pertaining to the energy contained in the oscillating part of the SLEM field is a measure of the mean energy lost due to the decoherence of free-electron motion. The corresponding part of the EM field is proportional to $\exp\left(i\omega_{l}-\Gamma_{l}^{coh}\right)\left(t\right)$ (Equation (Equation 10)), so:(11)$$Q_{l}^{coh}\left(R\right)=\frac{\omega_{l}\left(R\right)}{2\Gamma_{l}^{coh}\left(R\right)}.$$

However, the damping of the amplitude of the total freely evolving SLEM field causes, that the total quality factor of the SLEM field (Equation (Equation 10)):(12)$$Q_{l}^{tot}\left(R\right)=\frac{\omega_{l}\left(R\right)}{2(\Gamma_{l}^{pop}\left(R\right)+\Gamma_{l}^{coh}\left(R\right))}=\frac{Q_{l}^{coh}\left(R\right)}{3}$$
is smaller by the factor of 3: $\Gamma_{l}^{pop}+\Gamma_{l}^{coh}=3\Gamma_{l}^{coh}$. Size dependence of the quality factors for successive modes is presented in Figure 7 (left column).

### Optimization of the Quality Factors for Gold and Silver NPs with Respect to Spectral Range and Size

Radius dependence of the quality factors versus NP radius for gold and silver NPs are presented in Figure 7a. Their maximal values fall in the size range of relatively large NPs and increase with the mode multipolarity *l* (with exception of the dipole mode in Au NP) undergoing a shift towards still larger sizes.

In particular, the optimal radius $R_{l}^{opt}$ in the dipole mode (l= 1) is 23 nm for gold and 7 nm for silver NPs (see also Table 1). For still larger NPs, the plasmonic cavity in the dipole mode becomes weakly effective, as $Q_{l=1}\left(R\right)$ is a fast-decreasing function of *R* in that size range. However, quality factors for the consecutive modes with l>1 have their own maxima, with the optimal values for still larger radii. So, the optimal $Q_{l,R}$ can be obtained not in the range of smallest sizes, which is usually exploited in the experiments, but rather for relatively large nanoparticles for the above presently-available sizes, especially in the case of Ag NPs.

Figure 7b shows the dependence of the quality factors on the resonance frequencies $\omega_{l}$ of multipolar plasmon modes. The LSP resonance frequencies $\omega_{l}^{opt}$ corresponding to the maxima of the quality factors $Q_{l}^{tot}\left(\omega_{l}\right)$ with different *l* (and those of $Q_{l}^{coh}\left(\omega_{l}\right)$) are grouped in two spectral ranges: in the green in the case of gold and in the violet for silver NPs (see Figure 8) (see also Table 1).

Table 1 gathers the parameters $R_{l}^{opt}$ and $\hbar \omega_{l}^{opt}$, which are expected to optimize the performance of the plasmonic cavity in the dipole (l= 1) and the quadrupole (l= 2) modes for gold and silver NPs. The optimal resonance frequencies $\hbar \omega_{l=1,2}^{opt}$ can be excited with light wavelength $\lambda_{0}$ of the green and violet spectral range given in the right column of Table 1.

Let us notice, that the EM waves of frequency $\omega =\omega_{l=1}^{opt}$ (black, dashed lines in Figure 7) is not only quite effective in resonant excitation of higher multipolarity LSPs of the same radius as $R_{l=1}^{opt}$, but also for larger radii $R_{l}$ (Figure 7), since NPs excited at such frequency possess relatively large quality factors $Q_{l}\left(\omega_{l=1}^{opt}\left(R\right)\right)$. LSP excitation with the light wave of frequency $\omega \approx \omega_{l=1}^{opt}$ is expected to excite not only the dipole plasmon but also the higher multipolar plasmons. For the increasing size, $Q_{l=1}\left(R\right)$ diminishes. However, all the plasmon modes with l>1 are excitable for a given *R* until the decrease in $Q_{l}$ for the consecutive *l*. From a practical point of view, this effect is especially interesting in the case of Ag NPs, as relatively smaller Ag NPs form the effective plasmonic cavity at $\omega \approx \omega_{l=1}^{opt}$. The inset in Figure 7b shows the radii $R_{l}$ of Ag nanoplasmonic cavity with $Q_{l>1}$ as high as that of $Q_{l=1}\left(\omega_{l=1}^{opt}\left(R\right)\right)$. The corresponding quality factors are considerably higher than those for Au NPs.

However, the resonant plasmonic contributions to the measured intensities, associated with the TM modes of the EM field (described by the $B_{l}^{TM}$ coefficient of the Mie scattering theory [46]) are dominated by the contribution of the specular reflection [41] described by the terms containing a $B_{l}^{TE}$ coefficient. As the result, the resonant TM mode contribution, though important, can be obscured in the observation.

Let us also note, that in spite of the advantage of Au NPs as those possessing higher chemical inertness than Ag NPs, the quality factors for gold NPs are significantly smaller than those for silver nanoparticles in all consecutive modes and size ranges (Figure 8).

## 6. Conclusions

Understanding the properties of plasmonic systems and the application of their full potential to nanophotonic devices requires embracing their rich dynamic properties. One of the promising directions is the comprehension of plasmonic phenomena in nanostructures of larger sizes (radii of tens of nanometers or even larger), which are poorly explored. Knowledge of the exact size dependence of the multipolar properties of noble-metal nanoshperes, such as those of size-dependent resonance frequencies and damping rates (times), seems to suit that purpose.

The model of the plasmonic quasi-particle, complementary to the classical EM description allows finding the size dependence of the oscillatory dynamics of plasmonic EM fields and, above all, permits a clear distinction between the dephasing of the coherent behaviour of electrons involved in plasmonic oscillations and damping of the number of such electrons. It allowed us to clarify the meaning of the plasmon damping times, provisionally called $T_{1}$ and $T_{2}$, where $T_{2}$ is attributed to “the homogeneous line broadening” (e.g., [26,27,28,29,30,31] and references therein). Moreover, the applied model allows us to define such times for the case of multipolar plasmons described by the rates of populations and coherence damping, including their size dependence: $1/T_{1}\to \Gamma_{l}^{pop}\left(R\right)$ and $1/T_{2}\to \Gamma_{l}^{coh}\left(R\right)$, which is not monotonic. The derived radius dependencies allow flexible control of the dephasing times (especially in the case of silver NPs), contrary to some expectations (e.g., [31]).

Noble-metal nanostructures are most frequently used both in nanoscience and nanotechnology due to their superior plasmonic characteristics resulting from high optical conductivity and chemical inertness (especially gold) under ambient conditions. The size dependence of the quality factors $Q_{l}\left(R\right)$ for consecutive multipolar plasmons makes an objective tool for assessing the plasmonic performance of gold and silver nanoparticles. Due to the non-monotonic behaviour of both $\omega_{l}\left(R\right)$ and $\Gamma_{l}\left(R\right)$, the quality factors $Q_{l}\left(R\right)$ display characteristic maxima. It allows for optimizing the plasmonic performance of NPs in the spectral range from visible to near UV, by choosing the size of gold or silver NPs accordingly. All the parameters of NPs of practical interest (parameters of PI) (the radii *R*, the resonance frequencies related to the radii $\omega_{l}\left(R\right)$ and the resulting quality factors $Q_{l}$) are reflected in the EM near-field intensity and far-field absorption and scattering spectra, which display resonance-type behaviour. However, the manner they manifest in different measured quantities can be different. The expectations about the rules governing PI parameters manifestation in the measured quantities are usually based on the dipole-type model of a linear oscillator of self-frequency much smaller than the damping rate of oscillations (Lorentz profile, size effects not present). The present study of LSP intrinsic properties goes far above such approximation which is valid only in some ranges of parameters of PI presented in Figure 5, Figure 7 and Figure 8. In particular, it is shown that the values of IP parameters for the dipole mode in the lower limit of studied radii can be also found in a larger size range in the dipole mode, but also in the higher multipolarity modes. The expected LSP characteristics are nonmonotonic functions that can be interesting for different types of applications. In some applications, the radiative properties of a plasmonic nanoantenna are the most interesting. Such properties are expected in larger radius ranges, in regions where the rate of damping in consecutive modes increases with size and stays large. However, for some applications, it is crucial to have a sharp maximum on (or near) the LSP resonance frequency, and then the PI parameters are those corresponding to the maxima in the quality factors and take the values which are named optimal in the manuscript. In still other applications, the possibility of tailoring the plasmonic properties by size (or other PI parameters) is interesting. In that context, Figure 5, Figure 7 and Figure 8 deliver data about the ranges of PI parameters that strongly influence plasmonic characteristics.

However, AuNPs possess much poorer plasmonic properties than AgNPs. Additionally, the size dependence of the plasmonic resonance frequencies and quality factors for AuNPs is flatter than for AgNPs in the corresponding size ranges. That strongly favours silver NPs in applications (e.g., [7]), where tailoring plasmonic performance by engineering their size is of importance.

In addition, plasmons with higher multipolarity have interestingly large quality factors with maxima corresponding to larger NP sizes. As far as we know, this feature of NPs with larger sizes (over tens of nanometers in radius) has not yet been exploited experimentally. It is hoped that the rich dynamics of LSPs in spherical NPs covering a large range of sizes can also be helpful in understanding the plasmonic behaviour of larger plasmonic particles with non-spherical shapes.

## Figures and Tables

**Figure 1 materials-16-01801-f001:**
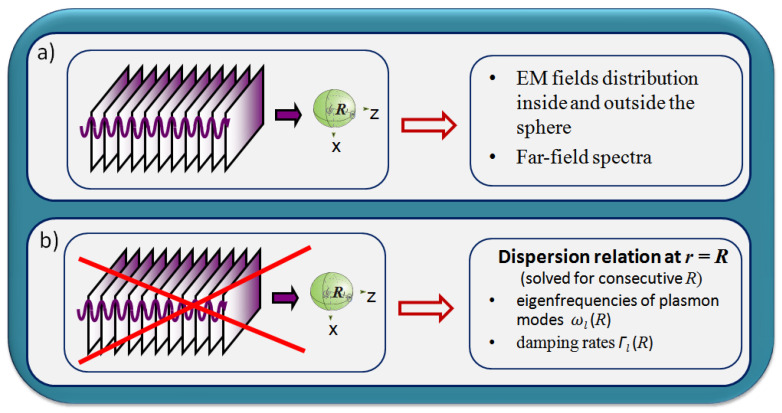
Illustration of the differences between the schemes used and the obtained data from (**a**) Mie scattering formalism (incoming field from external sources present) and (**b**) the self-consistent approach with no EM field present, which allows finding the dispersion relation at the spherical metal–dielectric interface.

**Figure 2 materials-16-01801-f002:**
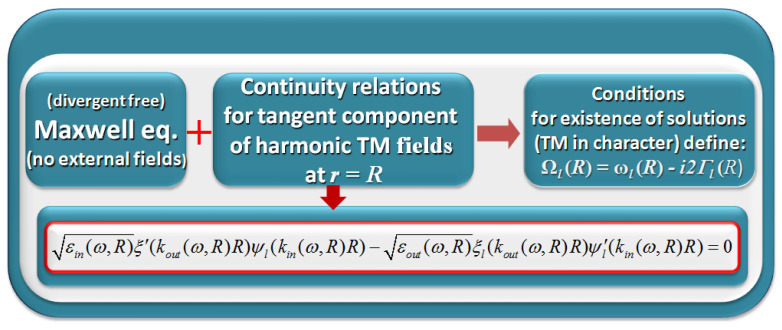
The scheme leading to the dispersion relation for TM surface-localized fields and the resulting conditions for the existence of solutions, which define the frequencies of the oscillations and damping of these oscillations for the consecutive modes l=1, 2, 3,…ψ and ξ are Riccati–Bessel spherical functions, the prime indicates differentiation with respect to the (complex) argument, $k_{in,out}=\frac{\omega }{c}\sqrt{\varepsilon_{in,out}}$.

**Figure 3 materials-16-01801-f003:**
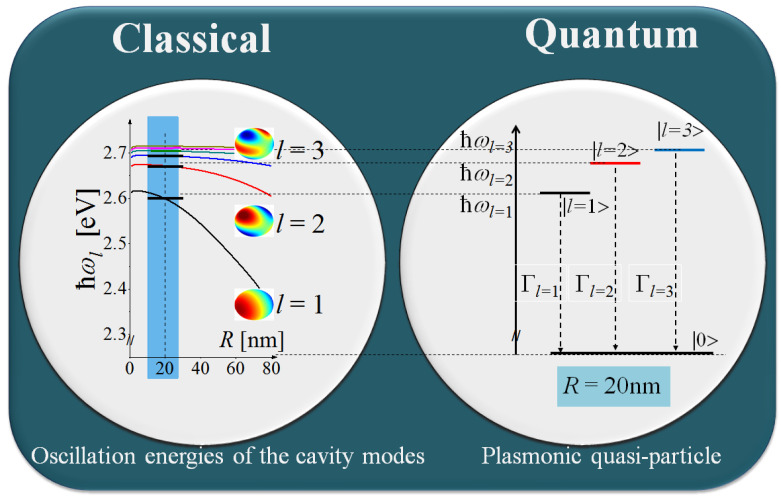
The scheme of relating the oscillation energies of LSP cavity modes to the energy levels of a plasmonic quasi-particle.

**Figure 4 materials-16-01801-f004:**
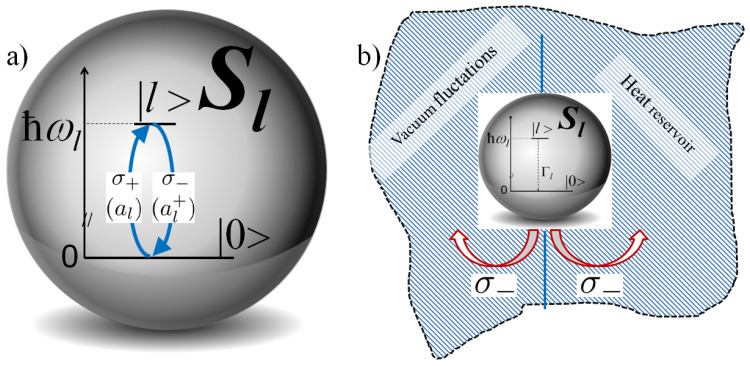
Illustrations of (**a**) two-level plasmonic QP and (**b**) irreversible coupling of a two-level plasmonic QP to the environment consisting of radiation and heat reservoirs.

**Figure 5 materials-16-01801-f005:**
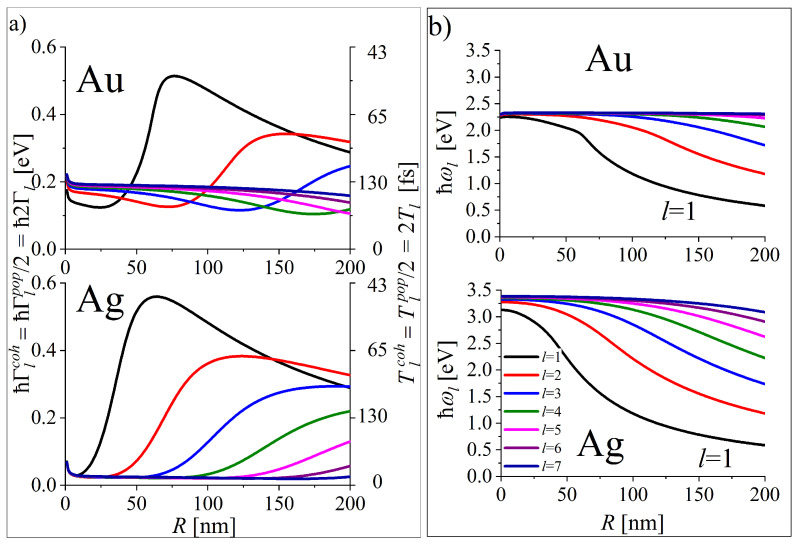
Size dependence of (**a**) the rates (left axis) of coherence damping $\Gamma_{l}^{coh}\left(R\right)$ and population damping $\Gamma_{l}^{pop}\left(R\right)=2\Gamma_{l}^{coh}\left(R\right)$ and corresponding times (right axis) (**b**) the energy levels of plasmonic QP (oscillation energies of SLEM fields resulting from the dispersion relation) for gold and silver NPs based on [15].

**Figure 6 materials-16-01801-f006:**
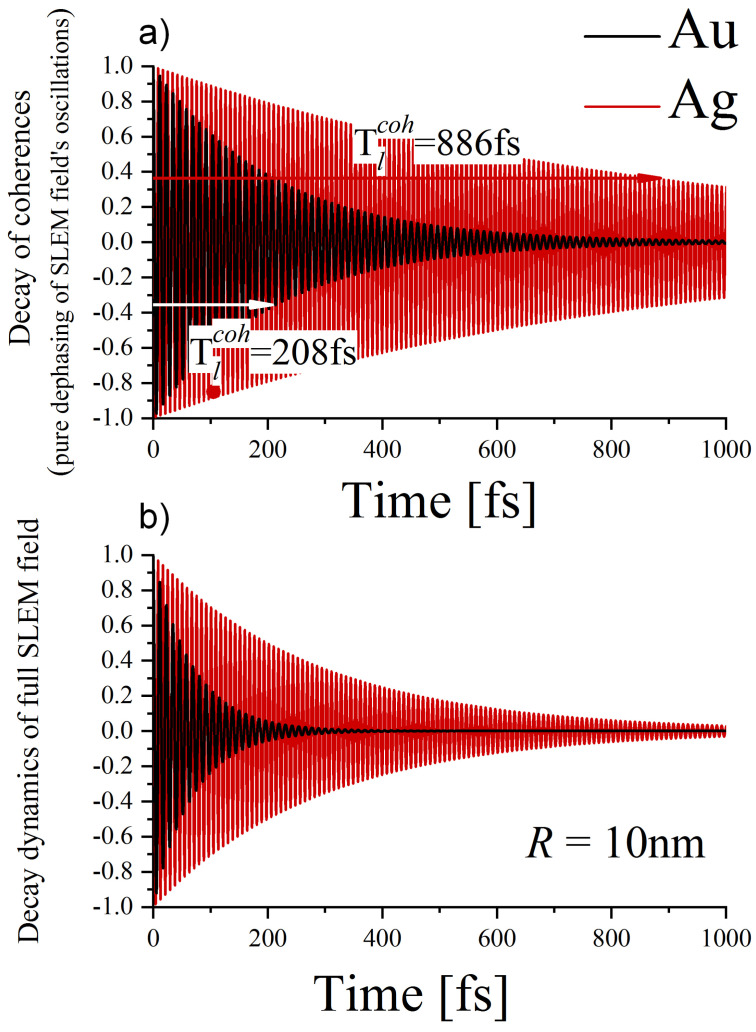
Illustration of the dynamics of the dipole-mode SLEM field produced by the free-electrons excited to the state l=1 for gold and silver NPs R=10 nm: (**a**) pure dephasing of the coherent oscillations (decay of coherence), (**b**) total dynamics of the SLEM field, which also includes the decay of the amplitude (of the population in the excited state).

**Figure 7 materials-16-01801-f007:**
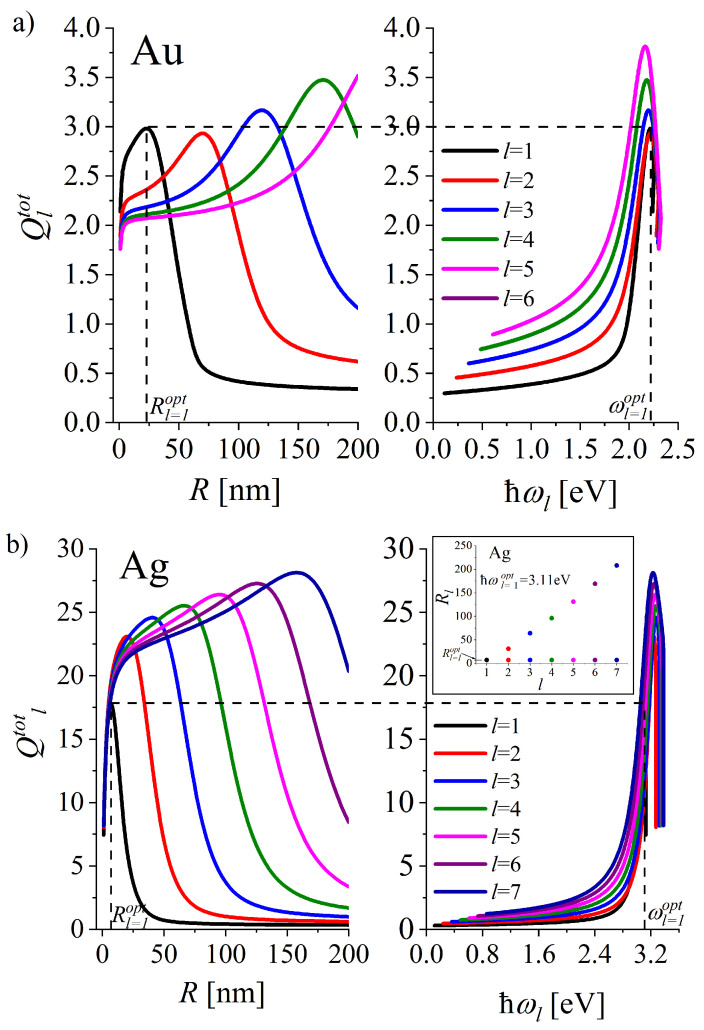
A diagram for optimisation of quality factors $Q_{l}$ of NPs of (**a**) gold and (**b**) silver. The left column shows the radius dependence, while the right column shows the corresponding dependence versus the resonant oscillation energy in the successive plasmon modes l=1,2,… 6. (see also Table 1). The inset shows the radii $R_{l}$ of Ag nanoplasmonic cavity with $Q_{l>1}$ as high as that of $Q_{l=1}\left(\omega_{l=1}^{opt}\right)$.

**Figure 8 materials-16-01801-f008:**
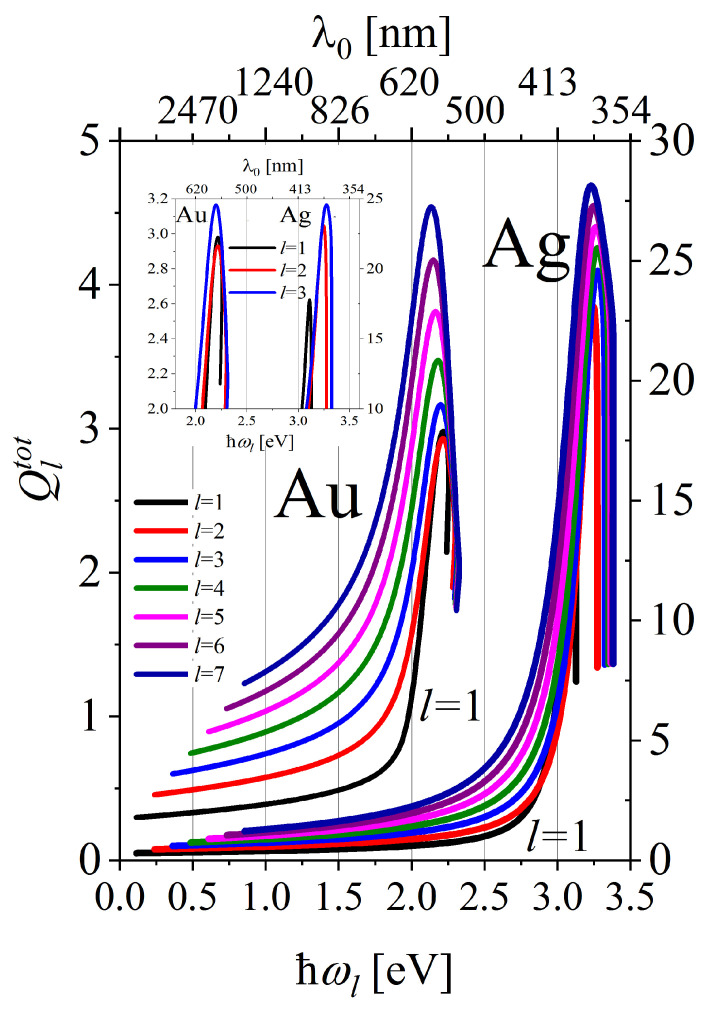
Quality factors $Q_{l}^{tot}$ versus LSP’s resonant oscillation energies $\hbar \omega_{l}$ for plasmon modes l=1,2…7 for gold (left ordinate) and silver (right ordinate) nanoparticles. The upper horizontal axis corresponds to the wavelengths of the incoming light wave enabling resonant excitation of plasmon modes. In the inset: the magnification of the first three LSP modes.

**Table 1 materials-16-01801-t001:** The optimal radii $R_{l}^{opt}$ and the optimal resonance frequencies $\omega_{l}^{opt}$ of NP cavity in the dipole (l= 1) and the quadrupole (l= 2) modes for gold and silver NPs in immersion oil ($n_{out}=$ 1.5), $\lambda_{0}$’s are the corresponding wavelengths enabling the resonance excitation of LSPs.

	*l*	$R_{l}^{\mathrm{opt}}$ [nm]	$\hbar \omega_{l}^{\mathrm{opt}}$ [eV]	$\lambda_{0}$ [nm]
**Au**	1	23	2.216	560 (green)
	2	70	2.16	560 (green)
**Ag**	1	7	3.11	399 (violet)
	2	20	3.24	383 (violet)

## Data Availability

Data available in a publicly accessible repository at doi.

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
