# Peer review of "Optimization of Coherent Dynamics of Localized Surface Plasmons in Gold and Silver Nanospheres; Large Size Effects"

_materials, 2023, doi:10.3390/ma16051801_

Round 1

Reviewer 1 Report

Combining results from the classical EM theory and the quantum Lindblad master equation, the authors obtained size dependence of the population and coherence damping rates for localized surface plasmons on nanospheres. The authors applied the theory to gold and silver nanoparticles. Their findings are illuminating. This work is well-written and educative. I recommend its publication in Materials. 

Author Response

Many thank for reading the manuscript. Please accept the amendments made (marked yellow in the corrected version). 

Reviewer 2 Report

The paper presents an interesting theoretical approach to fundamental calculations of spherical nanoparticles, especially of larger sizes, featuring multimode surface plasmons. The paper also attempts an equivalent quantum-mechanical treatment, based on the classical Mie scattering model and the two model are linked to provide a complete assessment of nanoparticle properties. 

1.The computational framework is only briefly described. It is understood that calculations are performed via Mie Theory but it is not clear how eigenfrequencies and eigenmodes are computed. The presentation of computational analysis need to be improved or complemented by proper references.

2. The introduction of the quantum mechanical model requires a better justification and a clarification as to whether it is actually involved in the computation or is basically a tool for better understanding of the underlying dynamics.

3. The paper requires also a better validation of results. The obtained high values of quality factors, especially in the case of silver nanoparticles pose a question whether metal losses in the optical regime have been correctly considered. This issue needs clarification.

4. Figures 1-3 are poster-like figures. It is suggested that dark background is replaced by plain white.

Author Response

I would like to thank you for reading the manuscript. Please accept the explanations,  and see the changes in the manuscript, which are marked as yellow.

The paper presents an interesting theoretical approach to fundamental calculations of spherical nanoparticles, especially of larger sizes, featuring multimode surface plasmons. The paper also attempts an equivalent quantum-mechanical treatment, based on the classical Mie scattering model and the two model are linked to provide a complete assessment of nanoparticle properties. 

1.The computational framework is only briefly described. It is understood that calculations are performed via Mie Theory but it is not clear how eigenfrequencies and eigenmodes are computed. The presentation of computational analysis need to be improved or complemented by proper references.

The method is based on solving the divergent-free Maxwell equations in the spherical coordinate system (that is the same as for the Mie scattering theory), but in absence of the EM wave coming from far sources. After applying the continuity relations for the TM modes, (such modes contain the normal to the interface component of the electric field), one can find the complex frequencies of modes when asking for the existence of solutions.

The method of finding frequencies of the mode oscillations and the rates of damping of these oscillations is described in Section 2, which contains two subsections: “Mie scattering theory versus dispersion relation (DR) for surface localized EM fields” and “Dispersion relation and the resulting oscillation energies and damping rates of plasmonic cavity modes versus metal NP's radius”. The Section is illustrated with two Figures: 1 and 2 and takes a little more than two full pages of manuscript. For more details the reader is invited to the references [47,48] as a main source of the dispersion relation derivation (lines 102, 124).

  1. The introduction of the quantum mechanical model requires a better justification and a clarification as to whether it is actually involved in the computation or is basically a tool for better understanding of the underlying dynamics.

The first idea is published in my paper in Plasmonics, referred to as [42].  In Section 3, which contains 3 Subsections, it is widely reconsidered, and, with Section 4 it forms a full description of the modelling I can presently offer. Please accept it..

  1. The paper requires also a better validation of results. The obtained high values of quality factors, especially in the case of silver nanoparticles pose a question whether metal losses in the optical regime have been correctly considered. This issue needs clarification.

Metal losses in gold and silver nanoparticles have been included in their dielectric functions (reference [15], published by me and my co-worker A. Derachova in Nanomaterials). This function includes ohmic losses, radius-dependent shortening of the free-electron length due to collision with the interface, and the impact of the interband transitions. If the Referee calls into question the validity of the results, more arguments are necessary.

  1. Figures 1-3 are poster-like figures. It is suggested that dark background is replaced by plain white.

                I am not against the use of Figures 1-3 in some posters or lectures if the paper is published and cited. The background is blue.

Reviewer 3 Report

1.     The abstract and introduction part is suggested to be well-organized to show the novelty of this work.

2.     What are the results of optimization of coherent dynamics of Localized Surface Plasmons in gold and silver nanospheres?

3.     What are the large size effects, and how big is the size of the particle?

4.     Why the comprehension of plasmonic phenomena in nanostructures of larger sizes is the promising direction.

5.     What is the exact size dependence of the multipolar properties of noble-metal nanospheres, does the small size and large size own the same rules?

6.     What is the difference between the small size and large size of plasmonic nanostructures in the application of generate, confine, guide, modulate and detect light?

7.     There are lots of spelling mistakes, such as:

1)     There is a dot in the Title, which is suggested to removed.

2)     Page 5, Line 171, “enery” should be energy

3)     Page 14, Line 319, “nanosheres” should be nanospheres.

4)     Page 14, Line 331,

5)     There are also lots of spacing between the words and the punctuations.

Author Response

Many thank for the effort in reading the manuscript and valuable suggestions for corrections. Please accept the explanations,  changes to the manuscript and amendments made according to your suggestions.

  1. The abstract and introduction part is suggested to be well-organized to show the novelty of this work.
  2. What are the results of optimization of coherent dynamics of Localized Surface Plasmons in gold and silver nanospheres?

All the parameters of NPs of practical interest (parameters of PI) (the radii, the resonance frequencies related to the radii and the resulting quality factors) are reflected in the EM near-field intensity and far-field absorption and scattering spectra. All these quantities display resonance-type behaviour. However, the manner they manifest in different measured quantities can be different.

The expectations about the rules governing PPI's manifestation in the measured quantities are usually based on the dipole-type model of a linear oscillator of self-frequency much smaller than the damping rate of oscillations (Lorentz profile, size effects not present).

As the paper goes far above such crude approximation, such expectations are a good approximation only in some ranges of parameters of PI which are available from the data presented in figures 5,7 and 8. E.g. it is shown that the values of IP parameters for the dipole mode in the lower limit of studied radii are also expected in a larger size range in the dipole mode and higher multipolarity modes and that the presented characteristics are nontrivial which can be interesting for different types of applications.

In some applications, the radiative properties of a plasmonic nanoantenna are the most interesting. Such properties are expected in larger radius ranges, in regions where the rate of damping in consecutive modes increases with size and stays large.

However, for some applications, it is crucial to have a sharp maximum on (or near) the LSP resonance frequency, and then the PI parameters are those corresponding to the maxima in the quality factors and take the values which are named optimal in the manuscript. In still other applications, the possibility of tailoring the plasmonic properties by size (or other PI parameters) is interesting. In that context, figures 5,7 and 8 deliver data about the ranges of PI parameters which strongly influence plasmonic characteristics.

A similar note is added to the Conclusions section, lines  342-374.

  1. What are the large size effects, and how big is the size of the particle?

By large size effects I mean the sizes still hardly available experimentally (explanation added to the Abstract, line 13 and page 2, line 64-65) and the text added in line 319: “radii o tens of nanometers or even larger”). The radius or the radii range are given in all Figures starting from Section 4 and extends to R=200nm. However, the data for larger radii can be calculated as well.

  1. Why the comprehension of plasmonic phenomena in nanostructures of larger sizes is the promising direction.

Please accept the amendment of p. 2.

  1. What is the exact size dependence of the multipolar properties of noble-metal nanospheres, does the small size and large size own the same rules?

All the parameters are calculated smoothly based on the same model, by changing R parameter by 1nm (no approximations concerning the radius, no formal distinction between small or large sizes).

  1. What is the difference between the small size and large size of plasmonic nanostructures in the application of generate, confine, guide, modulate and detect light?

By large sizes I mean the radii range hardly available experimentally (radius larger than tens of nanometers). I added the explaining words at the end of the abstract.

In the present stage of this study, the answer to this question can be only based on that given in point 2 (text added to the manuscript, lines 342-374. The more detailed answer is over the scope of the present study. I plan the next stage of the study which will include the deconvolution of the radiative and nonradiative (heat release) rates to the total multipolar damping rates in the function of radius which can give the background for a more complete answer to such kind of interesting problems, as divers applications require optimization of divers plasmonic properties.

  1. There are lots of spelling mistakes, such as:

1)     There is a dot in the Title, which is suggested to removed.                 Corrected

2)     Page 5, Line 171, “enery” should be energy                      Corrected

3)     Page 14, Line 319, “nanosheres” should be nanospheres.                Corrected

4)     Page 14, Line 331,                                            Reformulated

5)     There are also lots of spacing between the words and the punctuations.           Corrected

Reviewer 4 Report

I have go through this article " Optimization of coherent dynamics of Localized Surface Plasmons in gold and silver nanospheres; large size effects.", and found that overall the data provided looks good, it can be considered for publication in this journal. However, there are some issues that have to be fixed before publication;

The English language should be improved.

Moreover , There are so many errors in which need to be consider.

The title should be revised to somewhat catchy type.

In introduction section, there is need to modify by adding some text regarding the content by providing some new citations such as Applied Nanoscience 10, 1369-1378, Water 12 (2), 495.

The organization of the article is needed to change in a scientific way such in term of spacing problem, grammar and typo mistake etc.

Need to add more discussion on the obtained results by adding more references to support this study.

Some errors regarding the sub/super script, spacing and typo need to consider throughout the manuscript.

Make sure that the format of references are uniform.

Finally, a constructive and relevant conclusion is required.

Author Response

I have go through this article " Optimization of coherent dynamics of Localized Surface Plasmons in gold and silver nanospheres; large size effects.", and found that overall the data provided looks good, it can be considered for publication in this journal. However, there are some issues that have to be fixed before publication;

Thanks a lot to Reviewer 4 for going through the manuscript. I hope, you find my amendments satisfactory.

The English language should be improved. Moreover , There are so many errors in which need to be consider.

The errors I noticed have been corrected: lines 24, 172, 320, 331.

The title should be revised to somewhat catchy type.

                I have no idea how....

In introduction section, there is need to modify by adding some text regarding the content by providing some new citations such as Applied Nanoscience 10, 1369-1378, Water 12 (2), 495.

                Citations added:

                Applied Nanoscience 10, 1369-1378; added, [7], line 21, 58,367.

                Water 12 (2), 495; added, [35], line 58.

The organization of the article is needed to change in a scientific way such in term of spacing problem, grammar and typo mistake etc.

Corrected to some extent (marked as yellow).

Need to add more discussion on the obtained results by adding more references to support this study.

                Phrase reformulation: lines 323-324.

                Text added text to the Conclusions section, lines  341-359.

Some errors regarding the sub/super script, spacing and typo need to consider throughout the manuscript. Make sure that the format of references are uniform.

           Corrections: lines 290 and 320, double spacing was changed to single one.

References have been prepared using the “cite function” for  BibTex of the Google Scholar service, so in a uniform manner from my site. I would like to rely on this service assuming it is correct at this stage of the paper processing.

Finally, a constructive and relevant conclusion is required.

Text added to Conclusions section, lines  341-359.